# 3D finite element models reveal regional fatty infiltration modulates tibialis anterior force generating capacity in FSHD

Allison McCrady[1], Seth Friedman[2], Leo Wang[3], Dennis Shaw[4,5], Rabi Tawil[6], Jeffery Statland[7], Stephen Tapscott[3,8], Silvia Blemker[1]*

1 Department of Biomedical Engineering, University of Virginia, Charlottesville, Virginia, United States of America, 2 Center for Respiratory Biology and Therapeutics, Seattle Children's, Seattle, Washington, United States of America, 3 Department of Neurology, University of Washington, Seattle, Washington, United States of America, 4 Center for Clinical and Translational Research, Seattle Children's, Seattle, Washington, United States of America, 5 Department of Radiology, University of Washington, Seattle, Washington, United States of America, 6 Department of Neurology, University of Rochester Medical Center, Rochester, New York, United States of America, 7 Department of Neurology, University of Kansas Medical Center, Kansas City, Kansas, United States of America, 8 Fred Hutchinson Cancer Center, Seattle, Washington, United States of America

* ssblemker@virginia.edu

## Abstract

Facioscapulohumeral muscular dystrophy (FSHD) is a progressive neuromuscular disorder characterized by muscle damage, fibro-fatty infiltration, and ultimately weakness. The tibialis anterior (TA), very often involved relatively early in FSHD, is a primary dorsiflexor and important for ambulation. Recent work using magnetic resonance imaging to quantify fat infiltration in the TA volume observed a steep decline in force generation after fat reached ~20% in volume. Additional imaging studies have identified regional fat infiltration patterns that may contribute to the non-linear relationship between fat volume and muscle strength due to the distribution of fat within the muscle structure. The goals of this study were to 1) develop a pipeline for creating subject-specific models of the TA that include fat infiltration patterns measured from MRI and predict force generation, 2) compare models created using this pipeline with clinical measures of muscle strength, and 3) use the models to investigate the impact of regional fat distribution on muscle force generation. Twelve subject-specific models were created, and the model-predicted forces strongly correlate to clinical measures of strength in the same subjects (manual muscle testing (MMT): r = 0.75, and quantitative muscle testing (QMT): r = 0.54). The models showed fat amount accounts for 48% and muscle volume accounts for 74% of the variation in force. To investigate the impact of fat distribution, we developed eight pseudo maps to systematically vary fat location and amount in all subject-specific geometries. The models revealed that fat location modulates force generation, with the middle region involvement having the greatest impact in reducing force. This work highlights the need to characterize and understand the impact of intra-muscular fat distributions in neuromuscular diseases.

**Data availability statement:** All data needed to generate the models and replicate the results presented are available. The results of the model simulations are uploaded in supplemental information. The segmentations, voxel fat maps, contour points, model files, and project code are available on figshare at https://figshare.com/projects/3D_finite_element_models_reveal_regional_fatty_infiltration_modulates_tibialis_anterior_force_generating_capacity_in_FSHD/236591. The raw MR images are part of a larger NIH Wellstone study that is ongoing with a repository planned at study completion (2 of 5 years complete). Data-sharing requests regarding the NIH Wellstone study may be sent to Dr. Seth Friedman.

**Funding:** This work was funded by Friends of FSH Research (Dr. Friedman, co-PI), ARCS Scholar Award (Dr. McCrady, awardee), NIH T32 Biotechnology Training Program (T32GM136615, Dr. Blemker co-PI, Dr. McCrady, trainee), and NIH Awards (U01AR069393, Dr. Blemker, PI and R01AR078396-01A1, Dr. Blemker, PI). Data collection of original scans was conducted as part of the NIH Wellstone (P50 AR065139, Drs. Chamberlain and Tapscott). The funders had no role in study design, data collection and analysis, decision to publish, or preparation of the manuscript. URLS: NIH - https://reporter.nih.gov/ ; Friends of FSH - https://www.fshfriends.org/research/grant-history/building-mri-data-lake-potential-individual-muscle-quantification-and-data-aggregation-refine-biomarker-understandingsin-fshd; ARCS - https://www.arcsfoundation.org/scholars/danaher-foundation-grant-scholars.

**Competing interests:** The authors have read the journal's policy and the authors of this manuscript have the following competing interests: Silvia Blemker is co-founder and employee of Springbok Analytics and owns stock in the company. This does not alter our adherence to PLOS ONE policies on sharing data and materials.

## Introduction

Facioscapulohumeral muscular dystrophy (FSHD) is a neuromuscular disorder characterized by progressive fat infiltration that is associated with muscle weakness and functional decline [1]. FSHD muscle involvement is not uniform, often beginning in facial and shoulder girdle muscles, ultimately with pervasive weakness following decades of progression [2,3]. The tibialis anterior (TA) muscle is a common leg muscle showing early progression in FSHD [4]. The muscle dysfunction is a result of the toxic effects of mis-expression of the double homeobox 4 (DUX4) protein, including downstream dysregulated muscle repair, which results in progressive fatty infiltration of skeletal muscle observed across the body [5,6]. These structural changes to skeletal muscle result in loss of the force generating capacity of the muscle. Previous work has assessed this loss of strength through the correlation of total muscle fat infiltration by MRI, finding decreases in strength with increasing fat infiltration [7–9]. Interestingly, these studies have observed a non-linear relationship especially in the dorsiflexor muscle group [6,10], with a steep decline in force production above ~20% fat infiltration in the TA, a primary dorsiflexor [9]. This non-linearity is counter-intuitive to known structure function relationships of muscle, in which there is a direct proportionality between contractile muscle volume and force production [11,12]. This non-linearity suggests that the amount of intramuscular fat infiltration may not be the only driver of strength decline. Further characterization of infiltration patterns in relation to the muscle architecture and the impact on force generation must be understood to effectively assess the muscle strength of these patients.

The TA has a complex geometry with fibers originating from a proximal aponeurosis and the proximal third of the tibia and inserting into a central aponeurosis that becomes the long TA tendon [7,8,13]. Fatty disruption of this architecture may be key to understanding the non-linearity observed in fat infiltration to strength relationships in FSHD. Recent work using ultrasound imaging has identified reduced fascicle motion during contraction in FSHD, confirming fatty infiltration disrupts normal contraction mechanics in the TA [14], however this study did not quantify variations in the fatty infiltration patterns within the muscle volume. Indeed, recent work using magnetic resonance image (MRI) assessment of lower limb muscle involvement in FSHD has revealed a regionally heterogenous pattern of fat infiltration in the TA muscle [10,15]. These patterns show varying degrees of involvement from the distal to the proximal region of the muscle, with some primarily impacting the distal or proximal ends and others having a more homogenous fat infiltration along the length. These regional patterns involve varying disruptions of the complex fiber architecture of the TA, with distal and proximal infiltration specifically impacting fiber insertion and origin, which may explain the severe drop in force observed at low fat percent volume. However, revealing the primary source of force loss is challenging to uncover through experiments alone since multiple aspects of muscle architecture and composition change in FSHD.

Computational modeling provides a powerful approach for understanding muscle structure-function relationships and isolating the impact of individual factors on muscle force. Finite element modeling allows for complex geometries to be discretized

into elements to which mathematical constitutive material models can be assigned, and deformations can be simulated. Previous work has established a constitutive model of muscle, as nearly incompressible, hyperelastic, and transversely isotropic [16,17]. Additionally, this material model incorporates both active and passive properties of muscle, accounting for both along- and cross-fiber stretch and shear. Previous modeling efforts of fatty muscles have investigated intramuscular fat seen in sarcopenia [18]. This study demonstrated that simulation of 20% diffuse fat infiltration in a simplified gastrocnemius muscle showed a 45% reduction in force compared to a healthy muscle, as well as an increase in final fiber length, and decrease in fiber stress with increased fat. These models, however, were not image-based and the simulation of fat infiltration with diffuse distribution does not represent the complex heterogeneous fat infiltration patterns observed in FSHD. By leveraging artificial intelligence guided automated segmentation from MRIs [19], high fidelity subject specific geometries can be modeled with anatomic representations of individual fatty infiltration patterns. Additionally, finite element modeling allows for uncoupling of the complexities of the regional mapping of fat infiltration by systematic application of fat to the volume. The goals of this study were to i) develop a pipeline to rapidly create subject specific models of tibialis anterior that capture muscle geometry and fatty infiltration patterns, ii) compare model outputs to clinical measurements of dorsiflexion strength, and iii) evaluate the impact of regional distributions of fat on force loss. The tibialis anterior muscle was selected due to its common involvement, known variable fat infiltration patterns, and ease of comparing with clinical strength measurements.

## Methods

### MRI acquisition and segmentation

Twelve patients with FSHD (55.6 ± 15.3 years) who underwent T1 Dixon scans (3T Siemens scanner: TE1 = 1.35 ms, TE2 = 2.58 ms, TR = 4.12 ms, matrix = 448x266, voxel size = 1.1 × 1.1 × 4.0 mm, 104 slices) as part of a previous study were selected based on the fat infiltration and corresponding functional scores representing a spectrum of disease involvement (Table 1) [9]. The original study was conducted at the University of Washington, the University of Rochester, and University of Kansas through the Seattle Paul D. Wellstone Muscular Dystrophy Cooperative Research Center. The study was approved by the Human Subjects Committee at each institution, with written informed written consent obtained for all

**Table 1. Subject Demographics, clinical assessments of right dorsiflexor function, and TA measurements.**

| Subject | Sex | Age (yrs) | Height (cm) | Weight (kg) | Clinical Severity Score | Simplified Allele Length | MMT | QMT (N) | Muscle Volume (cm3) | Fat Fraction (%) | Muscle Length (cm) | Central Aponeurosis and Tendon Length (cm) | Internal Aponeurosis Length (cm) | Fascicle Origin Surface Length (cm) |
|---|---|---|---|---|---|---|---|---|---|---|---|---|---|---|
| 1 | M | 23 | 187.7 | 89.4 | 4 | 4 | 5 | 411.89 | 158.08 | 11.9 | 30.41 | 16.28 | 14.27 | 8.54 |
| 2 | M | 62 | 177.1 | 78.5 | 5 | 5 | 5 | 343.25 | 181.08 | 12.4 | 37.23 | 17.89 | 17.09 | 13.37 |
| 3 | M | 45 | 181.5 | 97.2 | 6 | 8 | 4.33 | -- | 115.32 | 13.5 | 34.41 | 20.16 | 18.08 | 10.73 |
| 4 | F | 61 | 175.1 | 86.2 | 6 | 5 | 5 | 60.80 | 96.91 | 13.9 | 29.88 | 17.18 | 15.58 | 7.97 |
| 5 | M | 30 | 180.0 | 77.1 | 5 | 2 | 2.67 | 24.52 | 93.77 | 14.3 | 29.78 | 17.75 | 16.55 | 8.60 |
| 6 | F | 65 | 171.5 | 54.8 | 5 | 4 | 4 | 362.86 | 73.31 | 21.9 | 28.11 | 15.31 | 13.19 | 8.31 |
| 7 | F | 70 | 170.0 | 100.7 | 3 | 6 | 4.33 | 372.67 | 103.25 | 34.2 | 27.13 | 15.78 | 13.78 | 7.47 |
| 8 | F | 68 | 156.3 | 43.0 | 9 | 6 | 3.33 | 196.14 | 46.86 | 38.4 | 26.03 | 16.14 | 14.53 | 8.90 |
| 9 | M | 65 | 182.9 | 94.9 | 6 | 8 | 5 | 117.68 | 169.24 | 39.1 | 28.24 | 16.31 | 14.31 | 8.99 |
| 10 | F | 53 | 167.0 | 69.4 | 5 | 5 | 3.67 | 19.61 | 55.37 | 49.4 | 26.01 | 14.84 | 13.60 | 8.80 |
| 11 | M | 61 | 174.0 | 98.0 | 8 | 2 | 2 | 26.48 | 102.57 | 65.4 | 29.02 | 16.57 | 15.37 | 8.46 |
| 12 | M | 65 | 171.4 | 81.0 | 8 | 9 | -- | -- | 90.95 | 68.5 | 29.19 | 16.95 | 15.19 | 10.05 |

Clinical Severity Score [21]; Simplified Allele Length [9]; MMT: Manual Muscle Testing [22,23]; QMT: Quantitative Muscle Testing [22,24,25].

participants. The de-identified imaging and segmentation data was accessed January 8, 2024 for this study. TA muscles were automatically segmented from axial slices using a combination of an artificially intelligent (AI)-based algorithm and manual vetting (Springbok Analytics, Charlottesville, VA, USA) [20]. Central aponeurosis and distal tendon were manually segmented from axial slices (3D Slicer) (Fig 1).

## Model geometry development

The segmentations were used to define contour points of the TA muscle volume, central aponeurosis to tendon, and tibia (MATLAB R2022b, Natick, MA, USA). The points were then used to define axial slice contours which were then lofted into the volumes of the TA and central aponeurosis (Autodesk Inventor 2024, San Francisco, CA, USA). The proximal apo-neurosis was simplified as a 15 mm extrusion of the most proximal axial muscle contour. The tibia points were imported to define the fascicle origin surface on the TA along the proximal third of the TA [7,13]. The volumes were meshed into tetrahedrons with the muscle-aponeurosis surfaces imprinted and merged (Coreform Cubit v2023.8, Orem, UT, USA). Fascicles and central aponeurosis fibers were defined using Laplacian fluid flow simulations and mapping the fluid velocity vectors to the mesh elements according to established methods (Autodesk CFD 2024, San Francisco, CA, USA) [26,27]. This method uses pressure driven flow by assigning boundary conditions of 1 Pa gage pressure to the fiber origin (inlet) surface and 0 Pa gage pressure to the fiber insertion (outlet) surface. The inlet surface of the TA was defined as the proximal aponeurosis and the surface defined from the tibia points, which ends at approximately 30% muscle length for all subject models (Fig 1). The outlet surface was defined as central aponeurosis. The inlet and outlet for the central apo-neurosis were the proximal and distal ends, respectively. All other surfaces were assigned a slip boundary condition. The volumes were meshed, and the velocity vectors for each element were mapped to the nearest element in the tetrahedron finite element mesh.

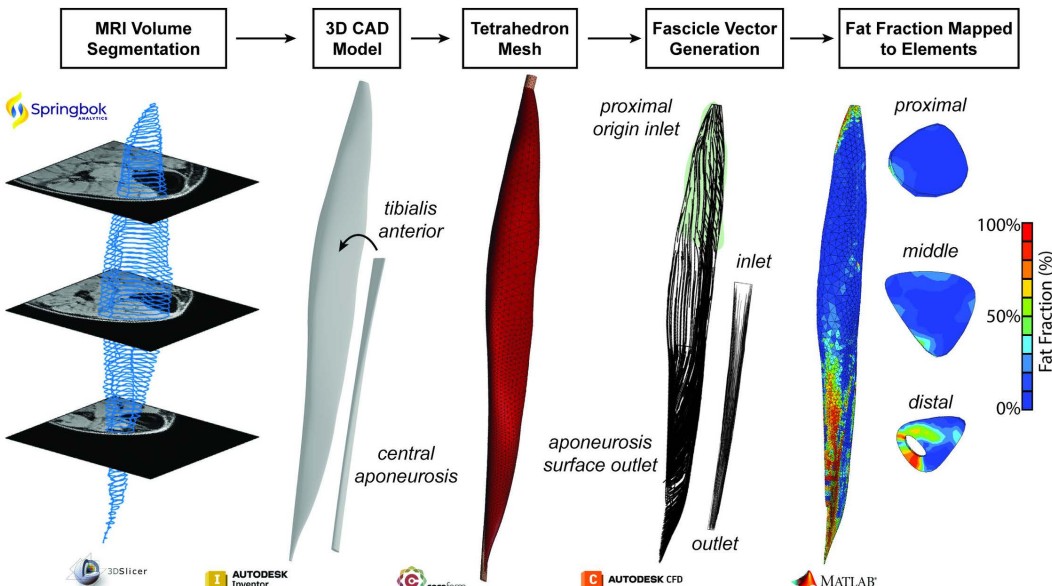

**Fig 1. Finite element model generation pipeline.** The finite element model geometry development pipeline begins with muscle volume and central aponeurosis segmentations from MRIs, followed by 3D computer aided design (CAD) models of the TA and central aponeurosis. These volumes were then meshed into tetrahedrons for FE analysis. The muscle fascicles and central aponeurosis fibers were generated through Laplacian fluid flow simula-tion and the velocity vectors were mapped to the mesh. Lastly, the voxel-based fat fraction determined from the Dixon MRI was mapped to each element in the volume, allowing for anatomic representation of the fat infiltration pattern.

## Fat mapping to muscle volume

The fat fraction for each voxel was mapped to the mesh, averaging the voxels within an element volume to calculate the associated fat fraction, and this is defined as the MRI-based map. From the MRI-based map, three additional methods of mapping fat to the muscle volume were implemented. First, a binary representation of the fat, with elements with greater than or equal to 50% fat fraction being assigned as fully fat, 100%, and all other elements were assigned 0%, or no fat. Next, the regional fat fraction for each third of the volume defined from the muscle length was calculated and mapped to all the elements in the region. The final anatomic-based representation of the fat was the total volumetric fat fraction mapped to all elements in the muscle for uniform fat distribution.

In addition to the anatomical maps, we developed pseudo maps that were uniform across all subjects to isolate the impact of regional fat infiltration on force loss. We generated eight pseudo mapping schemes that varied the average fat fraction within the specified region from 10–100% fat in 10% increments. The first three regions are the distal, middle, and proximal thirds of muscle length (Fig 2A). The fourth scheme simulates a common infiltration pattern identified clinically, with the distal and proximal ends infiltrated with fat (Fig 2B) [10], with the distal third and 20% of the proximal end infiltrated with fat. The remaining maps define the medial, lateral, anterior, and posterior regions of the volume by 50% width of each cross section along the respective medial/lateral and anterior/posterior axes (Fig 2C&D). To assign element-wise fat fractions, we utilized beta distributions to define a probability density function for each regional average fat fraction (Fig 3) [28]. To generate each distribution, we held $\alpha$ constant as 2 and varied $\beta$ to assign the desired average regional fat fraction ($\gamma$) as the peak of the distribution.

$$\gamma = \frac{\alpha}{\alpha + \beta}$$

(1)

Random fat fraction values were generated from the distributions and mapped to the elements. Each pseudo infiltration pattern was mapped to all 12 subject models, with 10 simulations per pattern to model the increasing fat content in the region.

## Material models and implementation of fat scaling

The muscle was modelled as an uncoupled solid mixture with muscle represented using a previously established transversely isotropic, nearly incompressible, hyperelastic constitutive model [16] and fat represented as an incompressible neo-Hookean material [18], and the ratio of these materials governed by the element specific fat fraction. The muscle material model includes the active and passive behavior of skeletal muscle. The strain energy density of the mixture is:

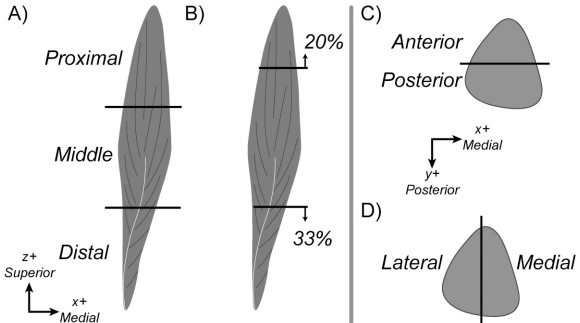

**Fig 2. Schematic demonstrating the eight pseudo mapping schemes showing the regions infiltrated with fat.** The figure shows the A) proximal, middle, and distal thirds, B) proximal-distal pattern, C) anterior and posterior regions in example cross-section, and D) medial and lateral maps.

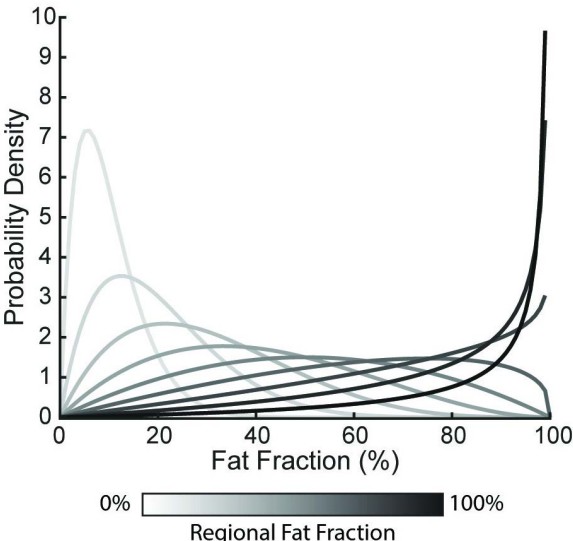

**Fig 3. Beta distribution probability density functions.** These distributions are used to apply fat fractions to each element for each pseudo mapping scheme. Darker lines indicate higher average regional fat fraction applied to the infiltrated region.

$$W_{total} = \left(1 - \frac{X_{fat}}{100}\right) W_{total}^{muscle} + \left(\frac{X_{fat}}{100}\right) W_{total}^{fat} \tag{2}$$

where $X_{fat}$ represents the fat fraction percent assigned to the element. The strain energy density of the muscle is defined as,

$$W_{total}^{muscle}(\lambda, \alpha, \varphi, \beta, J) = W_{total}^{iso}(\lambda, \alpha, \varphi, \beta) + W_{total}^{vol}(J) \tag{3}$$

where $\lambda$ represents along fiber stretch, $\alpha$ is muscle activation, $\varphi$ is along fiber shear, $\beta$ is cross fiber shear and $J$ is the relative change in volume. The deviatoric component is defined as,

$$W_{total}^{iso}(\lambda, \alpha, \varphi, \beta) = W_{\lambda}^{muscle}(\lambda, \alpha) + W_{\varphi}^{muscle}(\varphi) + W_{\beta}^{muscle}(\beta) \tag{4}$$

And,

$$\frac{\lambda \partial W_{\lambda}^{muscle}(\lambda, \alpha)}{\partial \lambda} = \sigma_{total}^{fiber}(\lambda, \alpha) \tag{5}$$

where $\sigma$ is the stress in the fiber, which is defined as,

$$\sigma_{total}^{fiber}(\lambda, \alpha) = \frac{\sigma_{max} * f_{total}^{fiber}(\lambda, \alpha) * \lambda}{\lambda_{ofl}} \tag{6}$$

Where $\sigma_{max}$ is the specific tension of the muscle, $f$ is the total force in the fiber summing active and passive forces [29], and $\lambda_{ofl}$ is the optimum fiber length. The along-fiber shear strain energy density is defined as,

$$W_{total}^{fat}(\varphi) = G_{\varphi}\varphi^2 \tag{7}$$

Where $G_\varphi$ is the along fiber shear modulus, and $\varphi$ is the along fiber shear deformation. The cross-fiber shear strain energy density is defined as,

$$W_\beta^{muscle}(\beta) = G_\beta \beta^2 \tag{8}$$

Where $G_\beta$ is the cross-fiber shear modulus and $\beta$ is the cross-fiber shear deformation. Lastly, the fat material was defined as a neo-Hookean material:

$$W_{total}^{fat}(I_1) = 2G_f(I_1 - 3) \tag{9}$$

Where $G_f$ is the shear modulus of the intramuscular fat, and $I_1$ is first invariant of the right Cauchy-green deformation tensor.

In this implementation, higher fat fractions will increase the contribution of the fat and reduce the contribution of the muscle material in the total mixture properties. The material parameters are defined from previous constitutive models of muscle [16,30,31] and are listed in Table 2, where K is the bulk modulus, $\varphi$ is the along-fiber shear modulus, $\beta$ is the cross-fiber shear modulus, $P_1$ and $P_2$ define the active and passive force-length curve, $\lambda_{ofl}$ is optimal fiber length, $\lambda^*$ defines the maximum fiber length at which active force is generated, and $\sigma_{max}$ is the specific tension. The aponeuroses were modelled as an uncoupled solid mixture with a Mooney-Rivlin ground matrix combined with fibers with an exponential power law [30]:

$$W_{total}^{apo} = W_{total}^{fibers} + W_{total}^{matrix} \tag{10}$$

Where the fiber material strain energy density is defined as,

$$W_{total}^{fibers} = \frac{ksi}{alpha(beta)}(\exp[\alpha(\lambda_{apo}^2 - 1)^{beta}] - 1) \tag{11}$$

Where alpha and beta define the exponential curve for the fiber behaviour, and ksi is the fiber modulus. The strain energy density of the Mooney-Rivlin ground matrix is defined as,

$$W_{total}^{matrix} = C_1(I_1 - 3) + C_2(I_2 - 3) + \frac{1}{2}K(\ln J)^2 \tag{12}$$

Where K is the bulk modulus, $C_1$ and $C_2$ are the coefficients of the first and second invariant terms of the right Cauchy-Green deformation tensor. All parameters are presented in Table 2.

**Table 2. Material parameters.**

| Muscle and Fat Material Parameters | | | | | | | | |
|---|---|---|---|---|---|---|---|---|
| K (MPa) | $\varphi$ (MPa) | $\beta$ (MPa) | $P_1$ | $P_2$ | $\lambda_{ofl}$ | $\lambda^*$ | $\sigma_{max}$ | $G_f$ (MPa) |
| 75 | 3.87E-03 | 2.24E-02 | 0.04 | 6.6 | 1 | 1.4 | 0.3 | 4.19E-03 |
| Aponeuroses Material Parameters | | | | | | | | |
| K (MPa) | $C_1$ (MPa) | $C_2$ (MPa) | alpha | beta | ksi (MPa) | | | |
| 500 | 5 | 0 | 0 | 2.5 | 1.5 | | | |

K: bulk modulus; $\varphi$: along fiber shear modulus; $\beta$: cross fiber shear modulus; $P_1$: exponential stress coefficient; $P_2$: fiber uncrimping factor; $\lambda_{ofl}$: optimal fiber length; $\lambda^*$: stretch at which the stress–strain relationship becomes linear; $\sigma_{max}$: specific tension; G: fat shear modulus; C1: coefficient of first invariant term; C2: coefficient of second invariant term; alpha: coefficient of exponential argument; beta: power exponential of argument; ksi: fiber modulus.

## Simulations and analyses

The boundary conditions were defined to simulate bony attachments during a fixed-end contraction. Zero displacement was prescribed in the x, y, and z directions at the distal and proximal ends of the aponeuroses as well as at the tibial origin surface (Fig 4). Muscle activation was ramped from 0 to 1 to model an active contraction. All simulations were performed quasi-statically in FEBio (FEBio Software, Salt Lake City, UT, USA) [32].

A sensitivity analysis on the force generation for different fat moduli was performed for three subject models as the stiffness of intramuscular fat remains to be characterized in FSHD. Fat shear modulus was varied from 1 to 15 kPa [33–35]. Model force output was then compared to manual muscle testing (MMT) scores and dorsiflexion quantitative muscle testing (QMT) values collected at the time of the MRI for each subject. Two subjects did not have a reported QMT, and one did not have a reported MMT. Pearson correlation was used to evaluate the relationship between model force output to clinical strength measures.

For each muscle geometry, a 'control' model with no fat material was created and used to simulate the maximal force potential for that muscle. The predicted force for each fat mapped model was then normalized to the force predicted by the associated control model to determine the percent of force generated relative the condition with no fat infiltration. The normalized force was then compared between the fat maps to examine how varying the distribution of the fat within the volume impact force generation. Next, we simulated ultrasound-like measurements from the model cross-section at 60% length from the distal end of cross-sectional area (CSA) and the average fat fraction within that section. These measurements were then compared to the MRI-based model force, allowing for investigation of how well cross-sectional area measurements represent the whole volume. Finally, the pseudo maps were used to modulate how the amount of fat versus where the fat volume is regionally infiltrated impacts force generation. The normalized force across each region and across the whole muscle were determined to examine how increasing fat in specific regions impacts force production.

## Results

Variation in the fat shear modulus values within the physiological range consistently led to no significant change in the force output across three models that had varying overall fat fraction levels (Fig 5).

There was strong agreement between the model force and manual muscle testing scores for the TA muscle (Fig 6A, r = 0.75**). There was moderate agreement between the model force and quantitative muscle testing (Fig 6B, r = 0.54). There were no significant correlations between model force and simplified allele length.

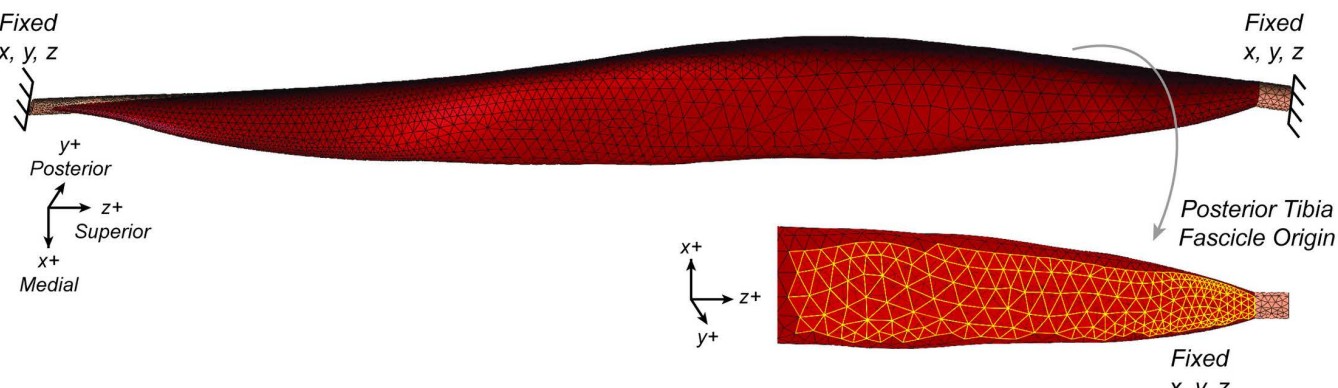

**Fig 4. Model boundary conditions.** This visualization is showing the fixed aponeuroses ends and fascicle origin surface from the tibia which is also fixed as a boney attachment.

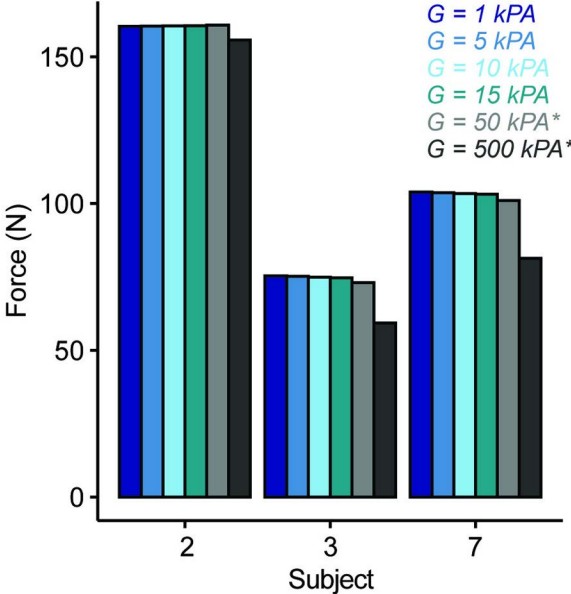

**Fig 5. Force sensitivity analysis.** The force output for varying fat shear modulus (G) for subjects 2, 3, and 7 between 1 and 15 kPa is shown. Additionally, two non-physiological values were tested (50 and 500 kPa) showing reduced force output with increased fat shear modulus. This analysis shows no sensitivity to the fat material properties for the force output within the physiological range.

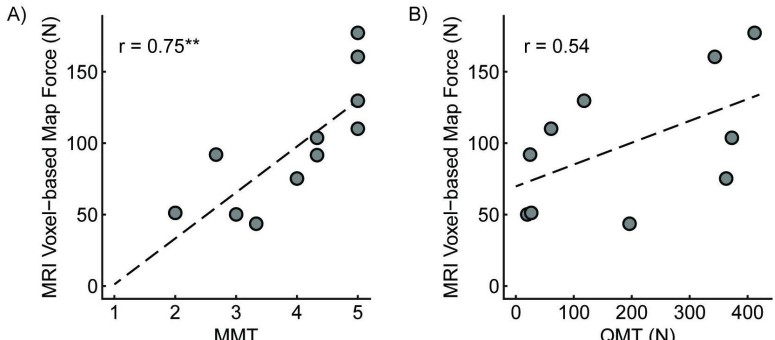

**Fig 6. Comparison of MRI-based model force to clinical strength assessments.** Pearson Correlations showed: A) manual muscle testing score (r = 0.75, p = 0.007), and B) quantitative muscle testing (r = 0.54, p = 0.1). Annotation: ** = p < 0.01.

Force loss differed across mapping schemes. There is a variation in fat infiltration patterns in the participant group (Fig 7).When observing the force for each model, there is a non-uniform force loss across the range of fat volume and distribution (Fig 8). For example, in the cases with higher fat fraction, there was greater force loss with the binary map compared to the MRI-based maps (Fig 9A&D). In the cases with lower fat fraction, there was more force loss in the other three anatomic schemes (MRI-based, regional, total) compared to the binary map. A linear relationship was observed between normalized force and volume fat fraction for the MRI-based (Fig 9A), regional (Fig 9G), and total (Fig 9J) map schemes. For all maps, when comparing force and volumetric fat fraction, there is a weak linear relationship, like that observed in previous work [9]. Lean muscle volume had a strong linear relationship ($R^2$ = 0.91, Fig 9C, F, I, L), demonstrating the volume of the contractile tissue is the main contributor to force generation.

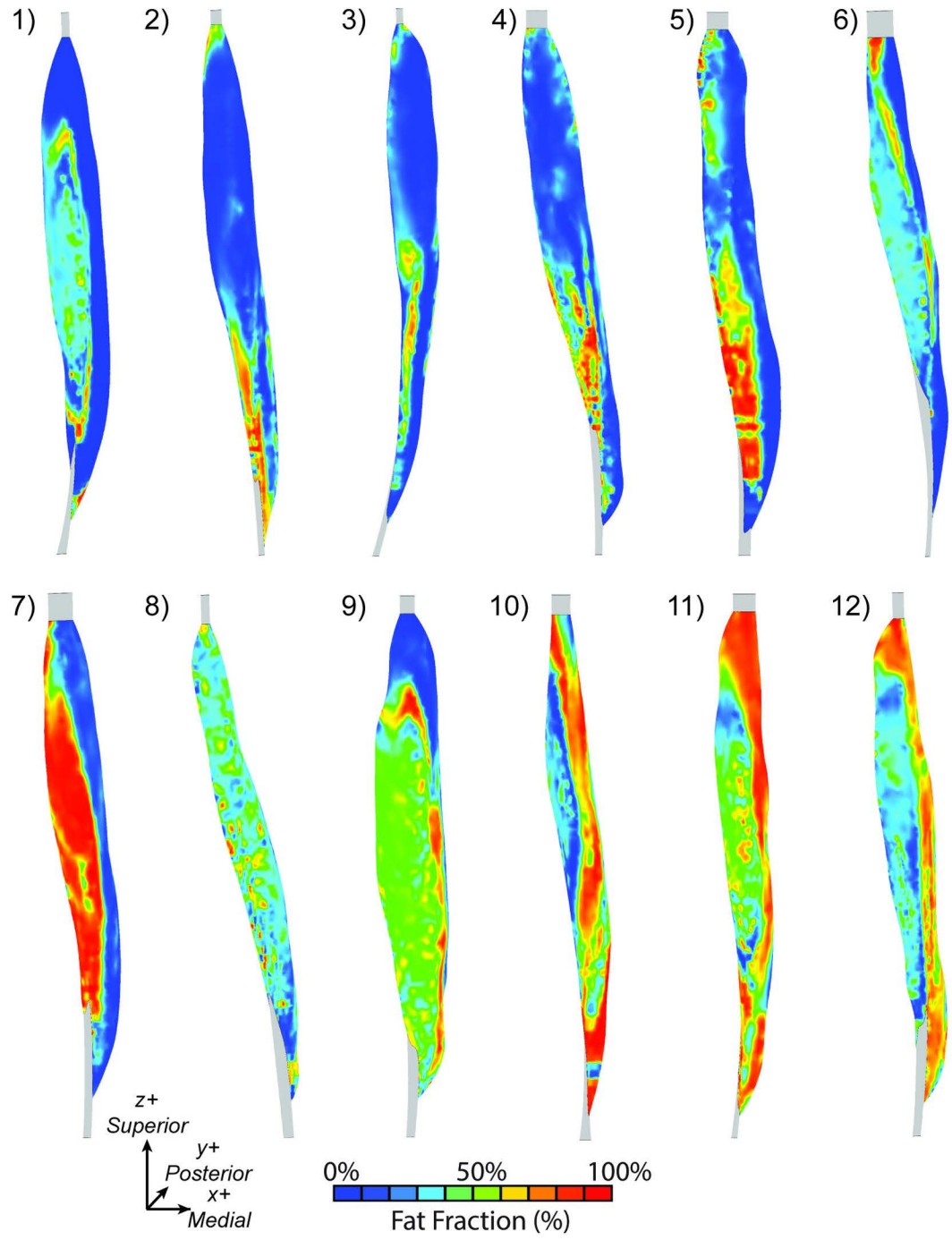

**Fig 7. 3D anterior view of the MRI-based fat fraction map.** This demonstrates the variation in fat infiltration patterns for all subjects. The subjects are numbered in order of increasing volumetric fat fraction.

Simulation of cross-sectional area measurements at the mid belly of the muscle indicate a strong positive relationship between CSA and force (Fig 10A, $R^2 = 0.71$), and a moderate negative relationship between mid belly fat fraction and force (Fig 10C, $R^2 = 0.42$). Interestingly, there are strong positive relationships between CSA at 60% length

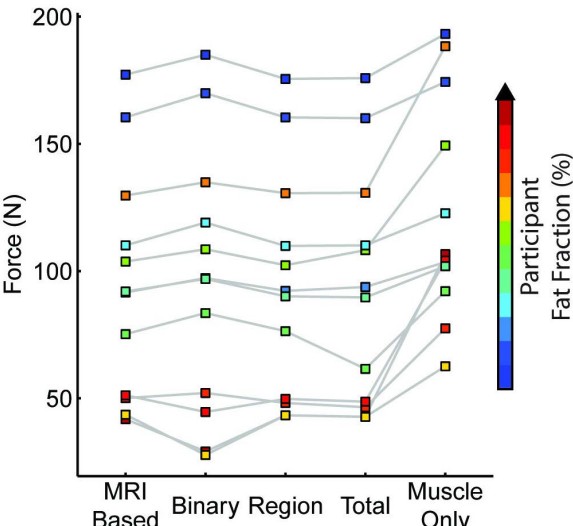

**Fig 8. Model force for each representation of anatomical fat distribution.** There is a non-uniform force loss across the range of fat volume and distributions modeled.

and muscle volume (Fig 10B, $R^2 = 0.91$) and mid belly fat fraction to volume fat fraction (Fig 10D, $R^2 = 0.94$) for all subjects.

Analysis of the pseudo maps shows a steep decrease in normalized force for fat infiltration in the middle region (Fig 11A). Distal end fat infiltration, including full fat replacement of the muscle volume, resulted in the smallest force loss across all pseudo maps. Comparing normalized force to volume fat fraction shows force loss modulation dependent on region infiltration in the TA. Additionally for a given volume fat fraction, for example 25%, each mapping scheme led to a different normalized force. The varying patterns in the superior-inferior direction had the most impact on the fat-fraction vs. force relationship (Fig 11A); however, varying patterns in the anterior-posterior and medial-lateral directions did not impact the fat-fraction vs. force relationship (Fig 11B&C).

## Discussion

In this study, we leveraged finite element modeling to generate subject specific models incorporating anatomic representations of volumetric fatty infiltration in patients with FSHD. The finite element model predictions reveal the impact of fat fraction on force generation from volume and architecture changes through comparison to a simulated healthy model, revealing a more linear relationship between fat fraction and normalized force. The developed pipeline allowed for rapid model creation empowering simulation of how fat infiltration patterns in the TA affected force production, which could be applied to other muscles and neuromuscular conditions.

We simulated four anatomically relevant mapping schemes and compared normalized force output. Indeed, we found increased fat fraction reduced normalized muscle force, consistent with known relationships between fat volume and muscle strength [15,36,37]. When evaluating the agreement of the predicted force to clinical function measurements, there was strong agreement with the MMT and moderate agreement with the QMT. This was expected as QMT measures force for a joint level dorsiflexor torque, along with additional normalizations applied that prevent it from being converted into an estimated muscle force for a more direct comparison. Interestingly, for the three subjects with between 10–15% volume fat infiltration, we saw less sensitivity to the mapping scheme, suggesting that the volume of muscle is dominating the behavior at these levels. However, the simplified representations of fat seem to have lower normalized force, therefore

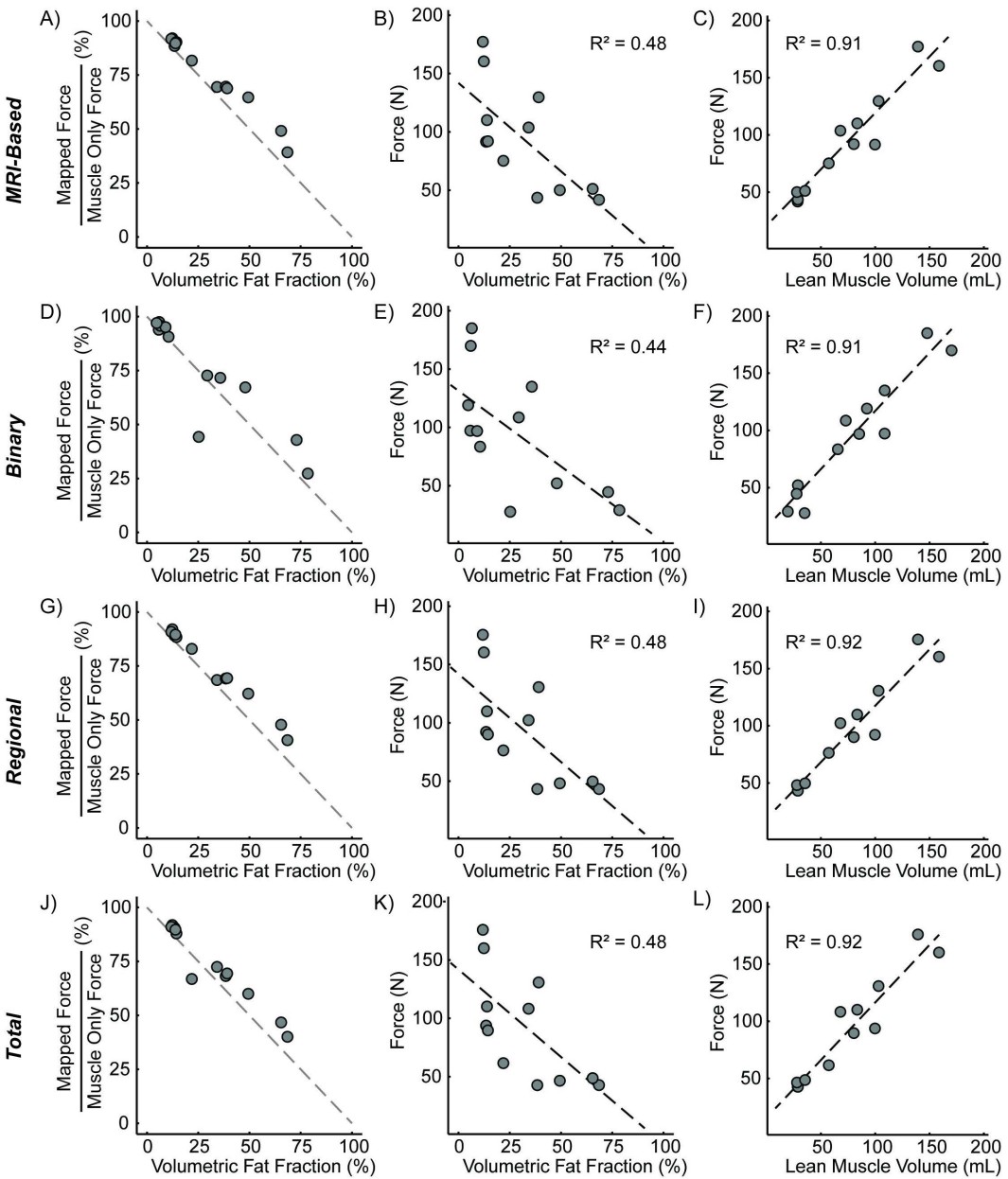

**Fig 9. Comparison of anatomic-based fat infiltration schemes.** MRI-based scheme – A) normalized force versus volume-based fat fraction, B) force versus volume-based fat fraction (this is the same as Fig 10F), and C) force versus lean muscle volume; Binary scheme – D) normalized force versus volume-based fat fraction, E) force versus volume-based fat fraction, and F) force versus lean muscle volume; Regional scheme – G) normalized force versus volume-based fat fraction, H) force versus volume-based fat fraction, and I) force versus lean muscle volume; Total scheme – J) normalized force versus volume-based fat fraction, K) force versus volume-based fat fraction, and L) force versus lean muscle volume. Gray dashed lines indicate a direct relationship between normalized force loss and volume fat infiltration. Black dashed lines indicate linear regression.

the fat may be modulating the muscle behavior due to larger regional concentrations of higher fat fraction values. For the subjects with higher volume fat fractions, we observed the simplified representations of fat infiltration in the regional and total maps seemed to over-estimate the normalized force compared to the binary map. Interestingly, we did not observe a non-linear relationship between volume fat fraction and normalized force in the anatomic-based maps as is reported in

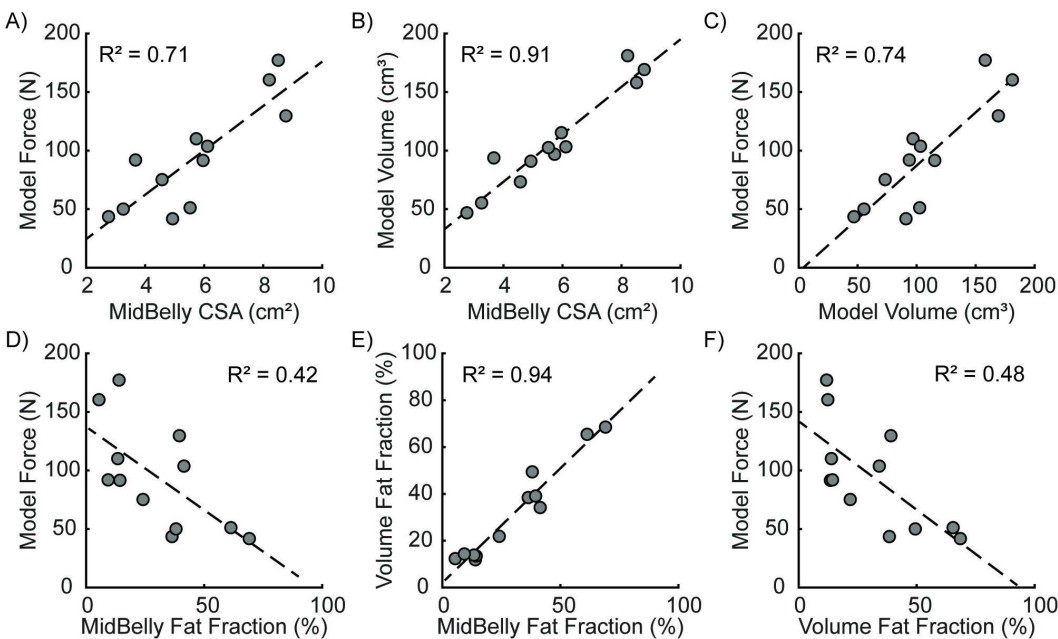

**Fig 10. Ultrasound measurement simulations.** The measurements are taken at the approximate mid belly (60% muscle length from distal end) with A) cross-sectional area (CSA) compared to force output, B) volume correlated to CSA, C) volume compared to force, D) mid belly fat fraction compared to force, E) mid belly fat fraction compared to volume fat fraction, and F) volume fat fraction compared to force (this figure is the same as Fig 9B).

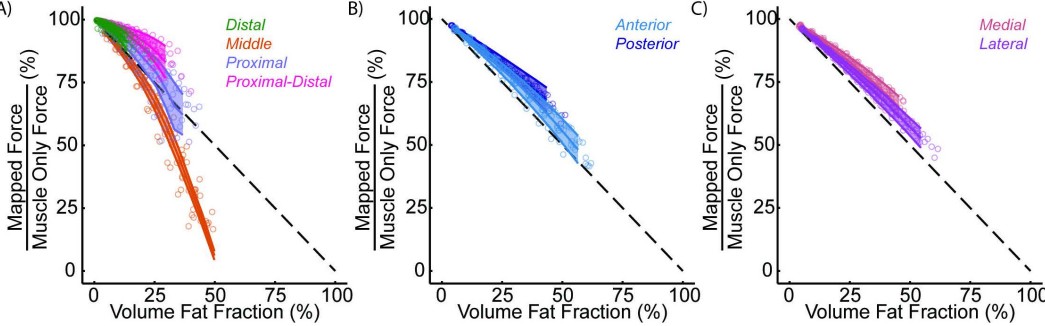

**Fig 11. Pseudo map normalized force output.** A) distal, middle, proximal, and proximal distal, and B) anterior and posterior, and C) medial and lateral mapping schemes. Dashed lines indicate a direct relationship between force loss and volume fat infiltration.

previous work [9,38], likely related to prior studies not separating the impact of muscle size and architecture changes from fat fraction. Indeed, our data suggests muscle atrophy and fat infiltration can happen in parallel and thus contribute to the non-linearity observed in real-world data. This suggests that by controlling for volumetric and muscle shape effects, fat infiltration has a direct impact on muscle force generation. Additionally, many of the TAs in this study showed higher fat infiltration along the medial side, which may also contribute to the linear relationship observed. Finite element analyses allow for isolation of the effect of increasing fat volume on force generation and confirm there is a direct effect of fat infiltration on force generation. Lean muscle volume seems to be a strong indicator of muscle force showing a strong positive linear relationship, consistent with recent work leveraging whole muscle analysis in Duchenne muscular dystrophy [39]. Future clinical work should consider calculating lean muscle volume in addition to fat fraction when assessing TA strength.

Many researchers are leveraging ultrasound imaging for the lost cost, portability, and ease of clinical use to quantify skeletal muscle changes during NMD progression [14,40–42]. However, ultrasound is limited by a small field of view and only captures a 2D representation of the muscle. By simulating cross-sectional area measurements (such as those collected from ultrasound) from our finite element models, we were able to compare the detailed 3D representation of muscle size and quality to 2D measurements. The CSA at 60% of the length of the TA was strongly related to muscle volume and force. This relationship is often observed in healthy muscles [12,43], however in NMDs such as FSHD with large increases in fat and fibrosis, there is often a weaker relationship between size and strength observed [44,45]. Ultrasound is often more variable due to user defined imaging planes, and therefore the CSA imaged may not be the maximum. We can better standardize our cross-sectional plane analysis in the model ensuring we are in the same region in each subject. Additionally, we observed the maximum CSA along the length of the muscle occurred at or near 60% of the muscle length for all our subject models, which may also help improve the strength of the size to strength relationship in these patient muscles. For the mid belly fat fraction, which would be similar to ultrasound measurements of echogenicity, we saw a negative relationship between force and fat fraction, consistent with ultrasound studies [46,47]. We observed a strong relationship between the fat fraction at 60% of the muscle and the total volume fat fraction. This may be due to the cross-section capturing all the functional compartments of the TA, however further studies on the anatomy of the functional compartments are necessary. More data is needed to fully determine the strength of relationship between 2D ultrasound and 3D MRI measurements of muscles in patients with FSHD.

In this study, we leveraged computational modeling to uncouple the relationships between muscle size and quality changes by applying pseudo maps to the subject volumes. This allowed us to systematically vary the fat location and amount while holding all other parameters constant and therefore elucidating the impact of regional fat distributions on muscle force generation. This analysis revealed that the regional location of the fat infiltration modulates the slope of the force loss with increasing fat volume. We identified the middle region of the volume was especially vulnerable, having the steepest decline in force generation with fat fraction. This result may be due to where the middle third of the volume is located relative to the fascicle architecture, and that this region includes both fascicle origin and insertion points, and that full fat infiltration in this region may completely disrupt whole fascicles. When considering the three compartments of the TA fascicle geometry first proposed by Wolf & Kim in 1997 [7], the middle region defined in this study encompasses all three compartments of the TA which may explain the steep loss of force generation. Further work is needed to characterize the impact of fat infiltration relative to the functional compartments of the TA and other muscles as well as on individual fascicle behavior. Interestingly, we observed minimal decline in force generation with increasing fat in the distal end and moderate decline in the proximal, proximal distal, anterior, posterior, medial, and lateral maps. This may suggest preservation of force generation until the fat propagates to the middle compartment and across the full cross-section. Indeed, although MRI studies conclude that the distal end may be the initiation point of fat infiltration propagation through the volume [10], we observed many anatomic fat infiltration patterns within the TAs in this study. The phenotype showing specific fat infiltration along the lateral side (Fig 7) may be an interesting clinical pattern and compartment to systematically model increasing fat infiltration. However, when applying fat to the lateral half of the TA, we did not observe a difference in force loss compared to the medial, anterior, and posterior halves, showing a near direct relationship to normalized force. This indicates it is important to characterize the fat along the length and across the CSA to fully understand how muscle strength may be impacted. Future work is needed to explore fat propagation along the muscle length and through functional compartments to elucidate regions in which fat infiltration is detrimental to force generation.

The development of a pipeline to rapidly model subject specific muscle geometry and fat infiltration patterns will be important for clinical translation of finite element analyses. The use of computational modeling, especially finite element modeling, allows for many simulations to inform the understanding of the clinical progression of FSHD, which would not be feasible in the real world. Additionally, we can expand these analyses to model fat infiltration patterns

across other muscles specifically effected in FSHD and incorporate other anatomical/functional landmarks to generate hypotheses about why fat deposition occurs in specific architectural regions. From these models, which can also be expanded also to muscle assemblies (e.g., extensors, flexors), there is the potential to more directly model more complex functional task performance being used in natural history and clinical trials. As a therapeutic biomarker, understanding how the location of fat infiltration within the structure of the muscle affects muscle force, we can improve the specificity of image-based clinical trial biomarkers. This will in turn bolster the use of image-based subunit parcellation assessments of muscles, having the potential to categorize and inform fundamental learning about the patterns of fat deposition seen in FSHD.

This study has several limitations to acknowledge. First, this is a subset of patients that may not fully represent the full spectrum of clinical fat infiltration patterns seen in the TA in FSHD. For example, some patients have dramatic atrophy of the TA with apparent hypertrophy of the extensor digitorum longus. On MRI this can be unambiguous to confirm but may be more difficult on ultrasound. Second, several simplifications to the geometry were made to expedite model simulations including the proximal aponeurosis and tibial attachment. Third, the central aponeurosis is essential to TA muscle structure and mechanics, and therefore further characterization of the aponeurosis behavior during contraction in these patients is necessary to understand how the aponeurosis may be altered in FSHD. Finally, we implemented a linear scaling of material parameters to model the fat distribution in the muscle, however further characterization of the interaction between fat and muscle tissue is necessary to improve the modeling of intramuscular fat.

In conclusion, this study revealed that the location of fat infiltration in the TA muscle volume modulates force generating capacity, with fat infiltration in the middle compartment being especially detrimental. While this work is specific to the TA, from an architecture perspective there are other muscles that should share similar features. Evaluating categories of muscles using similar methods will be the focus of future work. Generally, we believe this work highlights the need to characterize the regional distributions of fat within the muscle to improve image-based biomarkers for patients with neuromuscular diseases. Future work can leverage this finite element model development pipeline to explore the impact of fat on muscle fascicle behavior and using the pseudo map generation, systematically mapping clinical distribution patterns to the volume to generate additional insights on how the location of fat impacts muscle strength. The pipeline developed in this study is also applicable to other muscular dystrophies, especially Duchenne muscular dystrophy, where regional fat infiltration patterns have been identified [48,49], to improve the understanding of how fat impacts muscle force generation. In summary, this work developed a method to rapidly simulate fat infiltration patterns to improve the understanding of muscle weakness in FSHD, an approach that has broadly defined functional and mechanistic potential in future expanded muscles and cohorts.

## Supporting information

**S1 Fig. Average Fat Fraction of cross-sections along length for all subjects, highlighting the differences in the four variations of anatomical representations of the fat infiltration (MRI-based, binary, regional, and total maps).** S2 Fig.
(TIF)

**S2 Fig. 3D fat fraction maps for all subjects showing the aponeurosis footprint.**
(TIF)

**S3 File. Anatomical fat mapped model simulation results.**
(CSV)

**S4 File. Pseudo fat mapped model simulation results.**
(CSV)

## Acknowledgments

The authors graciously thank Springbok Analytics for the segmentations. The authors also thank the participants and families involved in this study.

## Author contributions

**Conceptualization:** Allison McCrady, Silvia Blemker.

**Data curation:** Allison McCrady, Seth Friedman, Leo Wang, Dennis Shaw, Rabi Tawil, Jeffery Statland, Stephen Tapscott, Silvia Blemker.

**Formal analysis:** Allison McCrady, Silvia Blemker.

**Funding acquisition:** Allison McCrady, Seth Friedman, Leo Wang, Dennis Shaw, Rabi Tawil, Jeffery Statland, Stephen Tapscott, Silvia Blemker.

**Investigation:** Allison McCrady, Silvia Blemker.

**Methodology:** Allison McCrady, Seth Friedman, Silvia Blemker.

**Project administration:** Allison McCrady, Seth Friedman, Leo Wang, Dennis Shaw, Rabi Tawil, Jeffery Statland, Stephen Tapscott, Silvia Blemker.

**Resources:** Allison McCrady, Seth Friedman, Silvia Blemker.

**Software:** Allison McCrady, Silvia Blemker.

**Supervision:** Seth Friedman, Silvia Blemker.

**Validation:** Allison McCrady, Silvia Blemker.

**Visualization:** Allison McCrady, Silvia Blemker.

**Writing – original draft:** Allison McCrady, Seth Friedman, Silvia Blemker.

**Writing – review & editing:** Allison McCrady, Seth Friedman, Leo Wang, Dennis Shaw, Rabi Tawil, Jeffery Statland, Stephen Tapscott, Silvia Blemker.

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
