## [Decision Letter · Decision Letter 0]

Dear Dr. Blemker,

Thank you for submitting your manuscript to PLOS ONE. After careful consideration, we feel that it has merit but does not fully meet PLOS ONE’s publication criteria as it currently stands. Therefore, we invite you to submit a revised version of the manuscript that addresses the points raised during the review process.

We look forward to receiving your revised manuscript.

Kind regards,

Atsushi Asakura, Ph.D

Academic Editor

PLOS ONE

Journal Requirements:

2**.** Please note that PLOS ONE has specific guidelines on code sharing for submissions in which author-generated code underpins the findings in the manuscript. In these cases, we expect all author-generated code to be made available without restrictions upon publication of the work. Please review our guidelines at https://journals.plos.org/plosone/s/materials-and-software-sharing#loc-sharing-code and ensure that your code is shared in a way that follows best practice and facilitates reproducibility and reuse.

“The authors have read the journal's policy and the authors of this manuscript have the following competing interests: Silvia Blemker is co-founder and employee of Springbok Analytics and owns stock in the company.”

4. In the online submission form, you indicated that “Data-sharing requests may be sent to Dr. Seth Friedman.”

Reviewers' comments:

Reviewer's Responses to Questions

**Comments to the Author**

1. Is the manuscript technically sound, and do the data support the conclusions?

Reviewer #1: Yes

Reviewer #2: Yes

2. Has the statistical analysis been performed appropriately and rigorously?

Reviewer #1: Yes

Reviewer #2: Yes

3. Have the authors made all data underlying the findings in their manuscript fully available?

Reviewer #1: Yes

Reviewer #2: Yes

4. Is the manuscript presented in an intelligible fashion and written in standard English?

Reviewer #1: Yes

Reviewer #2: Yes

Reviewer #1: The article, “3D finite element models reveal regional fatty infiltration modulates tibialis anterior

force generating capacity in FSHD”, is a hypothesis generating, technical article that aims to 1) develop a pipeline for creating subject-specific models of the TA that include fat infiltration patterns measured from MRI and predict force generation, 2) compare models created using this pipeline with clinical measures of muscle strength, and 3) use the models to investigate the impact of regional fat distribution on muscle force generation. The authors developed 3D finite element models (FEMs) to create subject-specific models incorporating fat infiltration patterns derived from MRI data. 12 subject specific models were created. They found that the model-predicted forces correlated cross-sectionally with clinical measures of strength (MMT>QMT) and that that fat amount accounts for 48% and muscle volume accounts for 74% of the variation in force. The creation of the 8 pseudo-maps enabled systematic variation of fat location and amount while holding all other parameters constant allowing for assessment of regional fat distributions on muscle force generation. They simulated different fat distributions (proximal to distal, anterior to posterior and medial to lateral) and found that fat location modulates force generation (mostly proximal to distal) with fat in the middle region having the greatest impact on reducing force. The authors also used ultrasound for 2D to 3D comparison to evaluate cross sectional area comparisons. Pearson correlation was used to compare model-generated force prediction with clinical strength tests (MMT and QMT) and sensitivity analysis was performed by varying fat shear modulus to assess its impact on force predictions.

The authors concluded that the FEM shows a more linear relationship between fat fraction and normalized force than previously published. They suggest that muscle atrophy and fat infiltration happen in parallel and therefore contribute to the non-linearity in real world data and that by controlling for volumetric and muscle shape effects, that fat infiltration has a direct impact on muscle force generation. While there is a direct relationship with fat fraction and force, it seems that there is a stronger relationship with lean muscle volume. The authors acknowledge this finding and suggest calculating both lean muscle volume in addition to fat fraction.

It is interesting that there was stronger agreement with MMT, a gross measure of muscle strength, rather than QMT which is thought to be more precise. They explain that this outcome was expected as QMT measures force for a joint level dorsiflexor torque, along with additional normalizations applied that prevent it from being converted into an estimated muscle force for a more direct comparison.

Questions for the authors:

1) In line 91, the authors mention that participants were chosen based on several different clinical features. Is it possible to include genetic characterization, e.g. number of repeats? If number of repeats are available, is there any correlation?

2) The TA was segmented with the used of AI and manual vetting. Would it be possible to automate the entire process?

3) Lines 225 – The authors describe how they predicted force. Does this take into account age?

4) In 230, the authors describe ultrasound measurements. I am not sure about the intent of the authors including the data from US as the relationship between the 2D US and 3D MRI. This data again seems to indicate that the better measure associated with force is muscle volume rather than fat fraction. The results from evaluation of the model as applied to US shows that CSA at 60% the length of the TA was strongly related to muscle and force. Are you implying that this measurement at 60% the length of the TA is a good assessment of the entire muscle volume? Could it be that US is a cheaper, easier method to assess association of fat replacement with muscle strength and potentially predict disease progression?

5) Ine 237 Why did variation of shear modulus in 3 subjects lead to no significant change? Is it because of the low fat fraction or because of lean muscle volume?

6) Please clarify if you will be able to apply this model to all muscles of the body or if additional models will need to be generated. You mention on line 363 and 389 that specific muscle geometry and fat infiltration patters will be important for all analyses. Can you provide an overview of which muscles would share similar features?

7) How will inclusion of lean muscle volume potentially change the model? It seems that there is a higher correlation of force with lean muscle volume than with muscle fat fraction.

They suggest that muscle atrophy and fat infiltration happen in parallel and therefore contribute to the non-linearity in real world data and that by controlling for volumetric and muscle shape effects, that fat infiltration has a direct impact on muscle force generation.

8) While there is moderate to excellent cross sectional correlation of fat fraction/location of fact fraction and lean muscle volume respectively, what parameters/clinical evaluations do you think will be important for assessment of longitudinal correlations?

Reviewer #2: This paper reports MRI-based 3D finite element models to show the relationship between fatty infiltration and force generation in TA muscles in FSHD. This study aimed to develop a pipeline for creating subject-specific models of the TA, to compare models created using this pipeline with clinical measures of muscle strength, and to use the models to investigate the impact of regional fat distribution on muscle force generation by twelve subject-specific modeling. This work is very interesting and logical for interpreting how fat replacement contributes to the decline in force generation. The introduction and discussion are well-written. However, there are difficulties to read.

1) At first, why was TA used for this study, not the facial, scapula, and humerus muscles?

2)Figure legends are not enough to explain the contents of Figures.

In Fig2, Fig3, Fig4, Fig5, Fig6, Fig8, only titles were provided. I could not get what is shown in Figure 8. Need the explanation for all Figs.

2) Figure 2 is interesting and should be moved to the Result section.

3) The authors said a more linear relationship between fat fraction and normalized force as shown in Fig 9. On the other hand, they concluded that the location of fat infiltration in the TA muscle volume modulates force-generating capacity, with fat infiltration in the middle compartment being especially detrimental, as in Fig11. However, fat distribution in TA is heterogeneous from patient to patient in Fig2. If the regional fat volume might have more effects on force generation, it should result in a non-linear relationship among the patients. Additionally, Fig10 shows that higher R2 was given by use of volume fat fraction, not midbelly fat fraction. Please explain that point.

4) Similarly, the relationship between Force and Lean muscle volume gave a high correlation coefficient R2>0.9. So total fat volume has more impact on force generation.

5) Please clarify that Fig10F is the same as Fig9B.

**Do you want your identity to be public for this peer review?** For information about this choice, including consent withdrawal, please see our Privacy Policy

Reviewer #1: No

Reviewer #2: No

---

## [Author Response · Author response to Decision Letter 1]

16 May 2025

We graciously thank the reviewers and editors for their time and effort in reviewing our manuscript and providing edits for our accepted manuscript. We have provided specific comments in the submitted "Response to Reviewers" document.

---

## [Editor Report · Decision Letter 1]

3D finite element models reveal regional fatty infiltration modulates tibialis anterior force generating capacity in FSHD

PONE-D-25-06669R1

Dear Dr. Blemker,

We’re pleased to inform you that your manuscript has been judged scientifically suitable for publication and will be formally accepted for publication once it meets all outstanding technical requirements.

Kind regards,

Atsushi Asakura, Ph.D

Academic Editor

PLOS ONE
---

## [Editor Report · Acceptance letter]

PONE-D-25-06669R1

PLOS ONE

Dear Dr. Blemker,

I'm pleased to inform you that your manuscript has been deemed suitable for publication in PLOS ONE. Congratulations! Your manuscript is now being handed over to our production team.

Kind regards,

on behalf of

Dr. Atsushi Asakura

Academic Editor

PLOS ONE